# Transcranial Direct Current Stimulation Modulates EEG Microstates in Low-Functioning Autism: A Pilot Study

**DOI:** 10.3390/bioengineering10010098

**Published:** 2023-01-11

**Authors:** Jiannan Kang, Xiwang Fan, Yiwen Zhong, Manuel F. Casanova, Estate M. Sokhadze, Xiaoli Li, Zikang Niu, Xinling Geng

**Affiliations:** 1College of Electronic & Information Engineering, Hebei University, Baoding 071000, China; 2Clinical Research Center for Mental Disorders, Shanghai Pudong New Area Mental Health Center, School of Medicine, Tongji University, Shanghai 200124, China; 3Department of Biomedical Sciences, University of South Carolina School of Medicine Greenville Campus, Greenville Health System, Greenville, SC 29605, USA; 4State Key Laboratory of Cognitive Neuroscience and Learning, Beijing Normal University, Beijing 100859, China; 5School of Biomedical Engineering, Capital Medical University, Beijing 100069, China

**Keywords:** autism spectrum disorder (ASD), transcranial direct current stimulation (tDCS), EEG microstate, autism behavior checklist (ABC) scale

## Abstract

Autism spectrum disorder (ASD) is a heterogeneous disorder that affects several behavioral domains of neurodevelopment. Transcranial direct current stimulation (tDCS) is a new method that modulates motor and cognitive function and may have potential applications in ASD treatment. To identify its potential effects on ASD, differences in electroencephalogram (EEG) microstates were compared between children with typical development (*n* = 26) and those with ASD (*n* = 26). Furthermore, children with ASD were divided into a tDCS (experimental) and sham stimulation (control) group, and EEG microstates and Autism Behavior Checklist (ABC) scores before and after tDCS were compared. Microstates A, B, and D differed significantly between children with TD and those with ASD. In the experimental group, the scores of microstates A and C and ABC before tDCS differed from those after tDCS. Conversely, in the control group, neither the EEG microstates nor the ABC scores before the treatment period (sham stimulation) differed from those after the treatment period. This study indicates that tDCS may become a viable treatment for ASD.

## 1. Introduction

Autism spectrum disorder (ASD) is a clinically and genetically heterogeneous neurodevelopmental disorder for which no single cause has been identified [1]. Studies have shown that ASD is related to multiple genetic and environmental factors [2,3]. According to the Diagnostic and Statistical Manual of Mental Disorders, Fifth Edition (DSM-5), the core symptoms of ASD are deficits in social communication, as well as restricted and repetitive patterns of behavior, interests, and activities. ASD is highly comorbid with other chronic medical disorders, including attention deficit disorder, sleep disorders, gastrointestinal disturbances, and movement disorders [4,5]. The Centers for Disease Control and Prevention (CDC) monitors epidemiological data on ASD to understand its impact on communities in the United States. According to these data, the prevalence of ASD is increasing, and the latest report published on 27 April 2018 indicated that the overall prevalence across 11 locations in the United States was 1 in 59 (approximately 1.68%). The CDC estimates that ASD is four times more likely to occur in males than in females [4,6]. Furthermore, most children with autism need lifelong care, and the lifetime expenditure for an individual with ASD with no intellectual disabilities is about USD 2 million to 2.4 million in the United States [7]. Although the literature indicates that treatment based on Skinner’s model of learning can improve ASD, no fully effective treatments have been developed [8,9].

Transcranial direct current stimulation (tDCS) is a noninvasive, low-cost, low-risk, and user-friendly brain stimulation technique [9] that is especially suitable for double-blind and sham-controlled experiments [10]. It transmits a 1–2 mA current through scalp electrodes to induce bidirectional, polarity-related changes in the cerebral cortex. Anodal tDCS can increase cortical excitability, whereas cathodal tDCS has an inhibitory effect [11]. The mechanism involves the facilitation or inhibition of synaptic transmission through an increase or a decrease in the action potential frequency of endogenous neuronal firing [12]. tDCS has measurable effects on neuropsychology and physiology [13]. A previous study provided the initial proof that tDCS could be used to improve mentalizing skills in persons with ASD traits [14], and it has been widely used in clinical research investigating Parkinsonism, major depressive disorder, and stroke [15].

Although the cause of ASD is unclear, research has suggested that dysfunction of the left dorsolateral prefrontal cortex (DLPFC) might play a role in ASD pathogenesis [16] and a study confirmed that observing actions involving implicit intentions most clearly revealed the impairment of the mirror neurons system (MNS) [17]. tDCS may ameliorate this dysfunction by altering neuronal and glial activity and synaptogenesis in the brain network and modulating brain metabolite concentrations (e.g., increasing the concentrations of N-acetylaspartate relative to creatine and myoinositol relative to creatine and decreasing the concentration of choline relative to creatine) in the left DLPFC [18]. These changes help elucidate the mechanism through which anodal tDCS improves cognitive processes associated with the DLPFC, including some of the potential effects of tDCS on ASD reported in previous studies [19,20].

Several studies have investigated the effects of tDCS in ASD by applying low-current stimulation (1–2 mA) to the left DLPFC for 20 min. For example, Amatachaya et al. found significant improvements in responses to the Autism Treatment Evaluation Checklist (ATEC) [21] after five sessions of anodal tDCS over the left DLPFC, including a significant decrease in the mean ATEC social subscale scores in the tDCS group [22]; in addition, Mahmoodifar and Sotoodeh found that combined tDCS and motor training enhanced balance in 6- to 14-year-old children with ASD [23]. Another study reported that 2 mA tDCS improved social cognition and skills in adults with ASD [24]. Furthermore, several studies investigating the potential of bifrontal tDCS as ASD treatment have suggested that bifrontal tDCS enhances working memory in adults with high-functioning autism [25] and has a moderate therapeutic effect on the behavior, sociability, and physical condition of children with ASD [26].

Guidance on the best outcomes to measure when applying tDCS in children with ASD lacks consensus. Electroencephalography (EEG) provides precise millisecond-scale temporal dynamics that measure the postsynaptic activities of cells in the neocortex, making it a powerful research tool for complex neuropsychiatric disorders [27]. The theory and methods of systems biology are important in the analysis of EEG because systems biology describes the brain as a complex biological dynamic system that can be described by two core concepts: state and dynamics [28]. The state of a system refers to the combination of variables needed to describe the system characteristics at a given time, and the dynamics of a system describes how the state changes over time.

EEG microstates are an important tool for studying the frequency and duration of various states or patterns of brainwave activity. EEG microstates were first proposed by Lehmann et al. [29]; they are hypothesized to be the foundational unit of neurological function and are proposed to represent an “atom of thought”. A microstate topology remains relatively stable over a period of 80–120 ms and then rapidly transforms into another topographical structure, resulting in a relatively stable topological pattern of microstates [30]. Researchers proposed that the stable period reflected simple information processing and was the basic unit of cognition, and they named this period the functional microstate [31]. Previous research indicated that four to eight microstate categories can explain approximately 64–83% of the variance in experimental data [32]. Research over the past few decades has determined that microstates A, B, C, and D are related to language processing, the visual network, the saliency network, and attention, respectively [33]. Topographic electrophysiological state source imaging was used to estimate the source of microstates [34]. Today, microstates are widely used to explore the pathological mechanisms of brain diseases [35].

Considering the findings discussed above, we analyzed EEG microstates and the Autism Behavior Checklist (ABC) scores [36] to examine the potential therapeutic effects of anodal tDCS over the left DLPFC in children, especially those with low-functioning ASD. We used EEG microstate analysis to investigate the changes in brain activity and the ABC scale to evaluate the changes in behavior, including sensory behavior, social skills, use of the body and objects, communication, and language [37].

## 2. Methods and Materials

### 2.1. Study Design

This randomized controlled trial enrolled 26 children with ASD and 26 children with TD. The 52 participants were divided into three groups: an experimental group (*n* = 13, 11 males, 6.52 ± 1.76 years), a control group (*n* = 13, 11 males, 6.38 ± 1.79 years), and a TD group (*n* = 26, 22 males, 6.64 ± 1.83 years). The study included three phases: (1) baseline evaluation, consisting of EEG analysis for all three groups and baseline ABC scale evaluation for the control and experimental groups; (2) a treatment period, in which 1 mA anodal tDCS was applied to the left DLPFC for 20 min twice a week for 5 weeks in the experimental group and sham stimulation was applied in the control group; (3) post-tDCS treatment evaluation, consisting of EEG analysis and ABC scale evaluation for the experimental and control groups. Parents examined and signed consent forms prior to their children’s participation, and they were informed that they could withdraw from the experiment at any time without consequences. Children with ASD continued their routine medication regimen and behavior training throughout the study. The study was conducted in accordance with the Declaration of Helsinki and was approved by the ethical committee of the Beijing Normal University.

### 2.2. Participants

Twenty-six children with ASD were recruited from three private centers for children with autism and randomly divided into two groups: a tDCS treatment (experimental) group and a sham stimulation (control) group. ASD diagnoses were confirmed by a child psychiatrist following a clinical review of the DSM-5 [38]. All of the children with ASD had low-functioning autism (IQ < 70). In addition, 26 children with TD whose ages and genders were matched with ASD children were recruited from kindergartens and primary schools. None of the participants had previously undergone brain stimulation treatment and none had a history of epilepsy, seizures, or severe neurological disorders (e.g., intracranial infections or brain tumors). Patients were required to report any notable symptoms during or after the tDCS or control procedures.

### 2.3. tDCS Treatment

A direct current of 1 mA was delivered via a battery-powered current stimulator (JX-tDCS-1, Huahengjingxing Medical Technology Co. Ltd. Nanchang, China). The impedance value between the two saline-soaked surface sponge electrodes (7 × 4.5 cm) was maintained below 50 kΩ during the stimulation process. The anodal electrode was placed over the DLPFC (F3), and the cathode electrode was fixed over the right supraorbital (Fp2) according to the 10–20 international system of electrode placement [39]. During the stimulation session, the current was ramped up from 0 to 1 mA over 30 s. Twenty minutes later, the current was ramped down from 1 mA to 0 over 30 s. The children in the experimental group (*n* = 13) received 10 tDCS sessions twice a week for 5 weeks. The tDCS device was also used for sham stimulation. The procedures for the control group were the same as those for the experimental group, but the current was discontinued after 30 s in the sham stimulation condition.

### 2.4. EEG Acquisition and Data Preprocessing

In the experimental group and the control group, EEG was conducted two times: once before any session and after all sessions of tDCS. In the TD group, EEG was conducted only once to compare EEG microstates between children with ASD and those with TD. EEG was collected in a soundproof and electrically shielded room by trained staff, and the participants were awake and in a resting state with their eyes open. Five-minute resting-state EEG was recorded with a 128 HydroCel Sensor Net System (Electrical Geodesics, Inc, Eugene, OR, USA) during the data-recording process; the impedance was controlled at less than 50 kΩ for all channels, and the sampling rate was 1000 Hz.

Matlab R2016a and EEGlab V13.5.4b were used for offline data analysis. Because the EEG signal was weak and easily influenced by external noise and experimental conditions, a 0.5–40 Hz band-pass filter was used to preprocess the EEG signal, which was segmented for a period of 4 s. Artifacts, including electromyogram (EMG), eye blink, and muscular artifacts, were manually removed by the same person following a specific amplitude and frequency threshold. According to the 10–10 international system of electrode placement, 62 electrodes were used for more accurate and faster results, and bad channels were interpolated. Finally, EEG data were re-referenced for average reference values. After preprocessing, the data were used for the next microstate analysis.

### 2.5. Microstate Analysis

In general, microstate analysis aims to identify the dominant template map from different scalp topographies in a given time domain. The analysis uses an appropriate clustering method to represent the transformation of the topographical structure of EEG data during a certain period and then calculates and marks the greatest correlation with the determined microstates at each time point of the original data. Finally, the desired temporal parameters are extracted from the fitted data. The differences in temporal parameters between groups can be determined for further analysis of their physiological significance. The temporal parameters calculated in this study were (1) the duration of the microstate; (2) the occurrence of the microstate; and (3) the time coverage of the microstate. These three parameters were used to evaluate the mean duration, occurrence frequency, and time coverage of potential large-scale brain networks during the resting state.

To obtain each participant’s dominant template maps, the global field power (GFP) was calculated for each sample time [40]:(1)$$GFP=\sqrt{\frac{\sum_{}^{}{\left(u_{i}-\bar{u}\right)}^{2}}{N}}$$
where $u_{i}$ is the voltage at an electrode, $\bar{u}$ is the average voltage of the electrode at the respective sample time, and N is the number of electrodes.

The scalp maps with the local GFP maximum at each sample time were submitted to an atomize and agglomerate hierarchical clustering (AAHC) algorithm to determine each participant’s dominant template maps [41]. Subsequently, the template map of each participant was extracted, and the AAHC algorithm was used again for a second clustering to obtain the microstate template map at the group level. The resulting template map was then fitted back to the individual EEG data in the time domain to define the microstate, and the temporal parameters were then calculated. Figure 1 shows the algorithm flow of EEG microstates.

### 2.6. Statistical Analysis

Statistical analysis was performed using the statistical package for social sciences (SPSS, version 20) [42]. Pairwise Wilcoxon signed rank tests [43] were conducted for each microstate in each temporal parameter to identify statistically significant differences. This is a common nonparametric test based on independent units of analysis for paired data (e.g., pre- and post-treatment measurements).

## 3. Results

The current study investigated the differences in EEG microstates between children with ASD and those with typical development (TD) and examined the effects of tDCS on brain activity and behavior in children with low-functioning autism. According to the metacriteria provided by CARTOOL [44], four microstate classes were determined to describe datasets at the group level by computing the global explained variance (GeV), which represents the number of microstate classes. When the cluster number was greater than four, the GeV showed almost no increase.

First, we constructed template maps and compared the differences in the four microstates between the ASD and TD groups. After fitting the four microstates back to the original data, we extracted the temporal parameters (duration, occurrence, and coverage) and calculated the significant differences. Figure 2 displays the differences in a template map across the two groups, and Figure 3 presents the differences in three temporal parameters of the four microstates.

Microstate B exhibited significant differences between the ASD and TD groups in all the three temporal parameters: duration (*p* = 0.0196), occurrence (*p* = 0.0005), and coverage (*p* = 0.0009). Microstate C showed statistically significant differences in duration (*p* = 0.0136) and coverage (*p* = 0.0204); microstate D exhibited significant differences in duration (*p* = 0.0025), occurrence (*p* = 0.0000), and coverage (*p* = 0.0000).

Next, we compared the template map differences of the four microstates, identified significant differences in the three temporal parameters, and calculated the changes between ABC scores before and after tDCS in the experimental group. Figure 4 presents the differences in the template maps, and Figure 5 illustrates the differences in the three temporal parameters of the four microstates before and after tDCS. Table 1 presents the ABC score changes in the experimental group before and after tDCS.

Compared with the pre-tDCS data, microstate A exhibited significant differences in duration (*p* = 0.0231), occurrence (*p* = 0.0062), and coverage (*p* = 0.0059) after tDCS; microstate C showed statistically significant differences in occurrence (*p* = 0.0057); and microstates B and D showed no significant differences. Significant differences were also observed in ABC scores before and after tDCS in social relating (*p* = 0.002) and language and communication skills (*p* = 0.019).

Finally, we calculated the template map differences of the four microstates and compared the differences between the three temporal parameters and ABC scores in the control group before and after tDCS, as shown in Figure 6 and Figure 7. No significant differences were observed in the three temporal parameters, and almost no changes in the ABC scores were observed in the control group.

## 4. Discussion

TDCS can target specific tissues and neural networks with minimal or no deleterious side effects for neurocognitive and behavioral functions. Anodal stimulation increases cortical excitability, whereas cathode stimulation inhibits it. These currents modify the transmembrane neuronal potential and thus influence the level of excitability, thus modulating the firing rate of neurons in response to additional inputs [45]. The depolarizing effects of anodal tDCS on neuronal resting membrane potentials and its demonstrated influence on LTP in neuronal circuits provide some explanation for the observed excitatory effects of anodal stimulation on behavior. Our previous study showed that the MER value was higher when comparing post-tDCS to pre-tDCS for the experimental group, which means that the EEG complexity increased after one session of tDCS stimulation [46].

In this study, we compared the microstates in resting-state EEG with the eyes open between patients with ASD and those with TD, providing insight into the aberrant intrinsic activities in the autistic brain. Microstate analysis was used to investigate the differences in three temporal parameters (mean duration, frequency of occurrence, and time coverage) between children with ASD and those with TD in resting-state EEG data. We also compared the changes in temporal parameters in children with ASD before and after tDCS to examine the effectiveness of tDCS. In addition, we calculated the changes in the temporal parameters in children with ASD who received sham stimulation, and no significant differences were found.

Previous studies have established the functional significance of microstates. Microstate A is associated with the negative blood oxygenation level-dependent (BOLD) activation in the bilateral superior lobe and the middle temporal lobe, which play key roles in speech processing [47]. Microstate B is related to negative BOLD activation of the bilateral occipital cortex, which is related to the visual resting network. Microstate C is associated with positive BOLD activation in the dorsal anterior cingulate cortex, the bilateral inferior frontal cortices, and the right insular area, which are related to self-representation and emotional communication [48]. Finally, microstate D is associated with the activation of negative BOLD in the right-lateralized dorsal and ventral areas of frontal and parietal cortices, which are related to the attention network [49].

First, the results illustrated decreased duration, occurrence, and coverage of microstate B in ASD compared with TD. Microstate B is thought to reflect the resting-state visual networks and is involved in imagery-related thoughts. A previous study suggested that atypical perceptual processing ability was associated with the autistic phenotype [50]. The fast extraction of the global gist was less efficient in ASD than in TD; by contrast, attention-driven processing of local elements was superior. Avraam et al. reported attenuated global processing in ASD [51]. A higher frequency of ASD occurrence has been reported in previous studies [52]. This might be due to the different EEG data obtained in the eyes-closed state. Few studies have been conducted on abnormal microstates in children with autism, and the age range was also a factor leading to different results.

Second, in the ASD group, the duration and coverage of microstate C, which is related to self-representation, were significantly decreased compared with those in the TD group. One of the characteristics of ASD is the impairment of social communicative development. Our results indicate that these impairments may be due to the premature termination of cognitive processes related to self-representation in children with autism, which means that they do not have enough time to fully process relevant external or internal information.

Third, the duration, occurrence, and coverage time of microstate D were significantly higher in the ASD group than in the TD group, which suggests that the underlying ventral attention network (VAN) related to microstate D in the ASD group was different from that of the TD group [53]. VAN is responsible for nonspatial attention, including event awareness, attention reorientation, and alertness, which are typically abnormal in the behavior of children with ASD [54]. It was also revealed that the VAN region was located in the right frontal lobe, especially the right middle frontal gyrus [55]. Previous research revealed abnormalities in the frontal regions in ASD [56], and our results confirmed these findings.

This study also investigated the effects of tDCS treatment on ASD. In the experimental group (children with ASD who received tDCS), we calculated three temporal parameters of the four microstates before and after tDCS. The duration, occurrence, and coverage of microstate A after tDCS were significantly higher than those before tDCS. The increases in microstate A parameters indicate that the BOLD activation in the bilateral superior lobe and middle temporal lobe and the phonological processing time increased after tDCS treatment. The ABC scores, namely social relating and language and communication skills scores, also significantly improved, which suggests that tDCS has strong effects on ASD. We did not observe any significant differences in microstates and scale scores in the control group.

The current study differs from previous studies in three keyways. Firstly, the subjects of most previous studies on EEG microstates in ASD have been adults or patients with high-functioning autism; no similar studies have included 3- to 6-year-old children. Second, many previous studies have used eyes-closed, resting-state EEG signals, whereas we used eyes-open, resting-state EEG signals. It would have been very difficult for children with low-functioning autism to keep their eyes closed for the whole experiment. Therefore, the original EEG data contained a high amount of interference, which might have resulted in inaccurate results. Previous studies have shown that microstates are more relevant to the alpha frequency band, and the eyes-closed data can better reflect the alpha frequency band activity [57]. However, we tried to collect the eyes-closed data of autistic children, but failed. Our previous studies also showed that the eyes-open data can also reflect the abnormalities of psychiatric diseases [58]. Third, previous studies on the effects of tDCS on autism have mostly been based on differences in scale scores, which tend to be subjective. In this study, we included more objective outcomes by combining ABC scores with EEG microstate analysis.

## 5. Conclusions

Microstates A, B, and D differed significantly between children with TD and those with ASD. tDCS displayed significant differences in EEG microstate and ABC scale between pre- and post-tDCS in the experimental group. Conversely, in the control group, neither the EEG microstates nor the ABC scores before the treatment period (sham stimulation) differed from those after the treatment period. This study indicates that tDCS may become a viable treatment for ASD.

### Limitations

The current study analyzed EEG microstates and ABC scores, and the findings suggest that tDCS can have positive effects on children with low-functioning autism; however, the study has two major limitations. The first limitation is that EEG data were collected in the eyes-open condition; this condition was selected for the convenience of the participants, but it may have interfered with the microstate parameters. The second limitation is the relatively small sample size. Our future studies will enroll more participants and investigate whether gender has any effect on the time parameters of microstates.

## Figures and Tables

**Figure 1 bioengineering-10-00098-f001:**
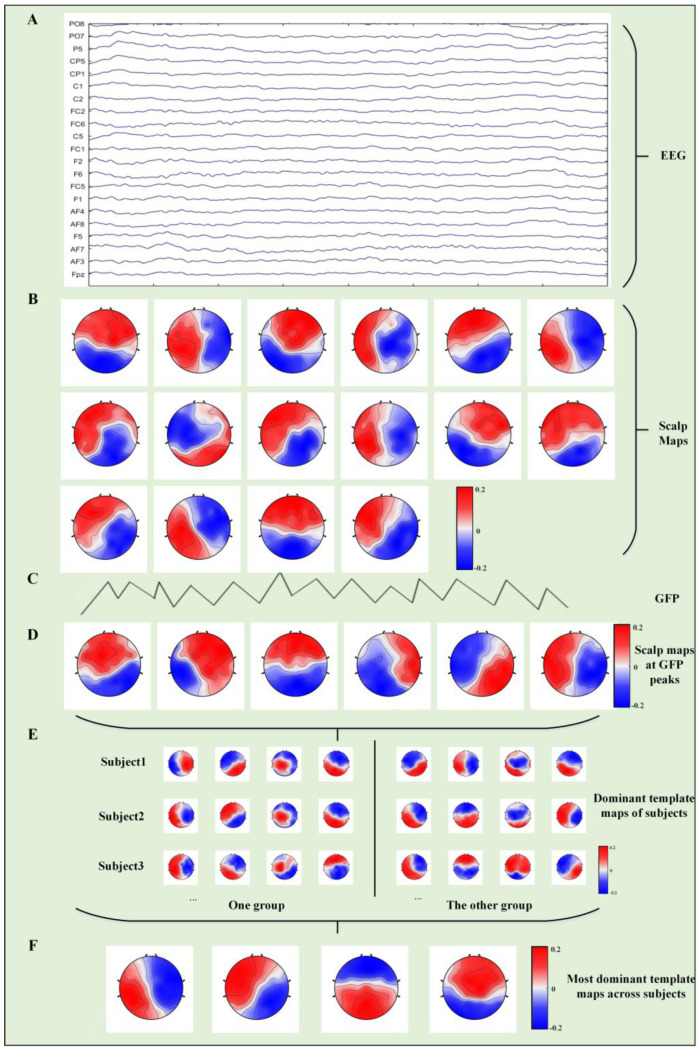
Electroencephalogram (EEG) microstate analysis. (**A**) Original EEG data. (**B**) Sixteen scalp maps at 0.5 s intervals. (**C**) Identification of the local peaks at the global field power (GFP) curve within 0.5 s intervals. (**D**) The scalp maps corresponding to the local GFP peaks were submitted to an AAHC analysis. (**E**) The dominant template maps for each participant. (**F**) The dominant template maps for the four microstates after group clustering across all participants.

**Figure 2 bioengineering-10-00098-f002:**
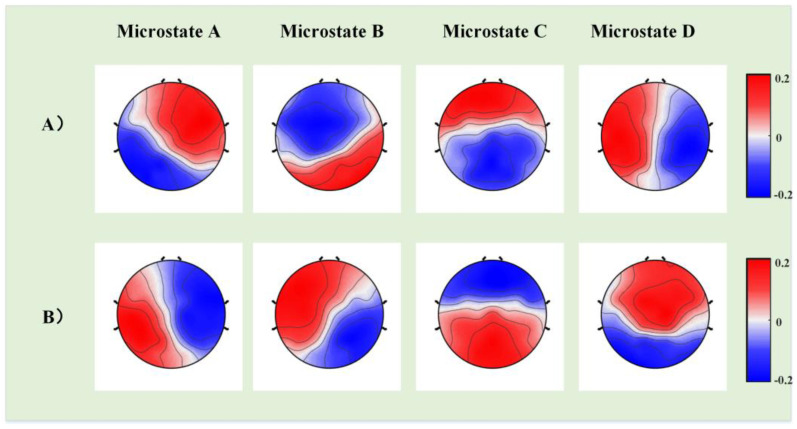
The group-level template maps of the four microstate classes. (**A**) The microstate template map in the autism spectrum disorder (ASD) group (*n* = 26). (**B**) The microstate template map in the typical development (TD) group (*n* = 26).

**Figure 3 bioengineering-10-00098-f003:**
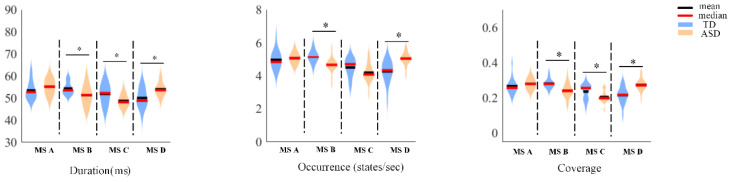
The differences in three-time parameters of the four microstates between the ASD and TD groups. * denotes *p* < 0.05.

**Figure 4 bioengineering-10-00098-f004:**
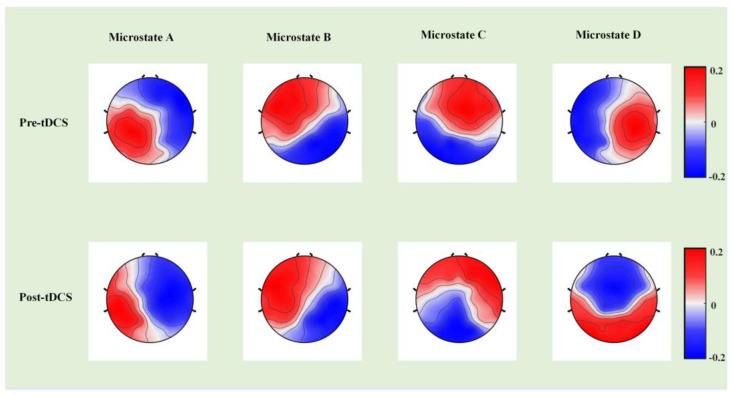
The template maps of the four microstate classes in the experimental group before and after tDCS.

**Figure 5 bioengineering-10-00098-f005:**
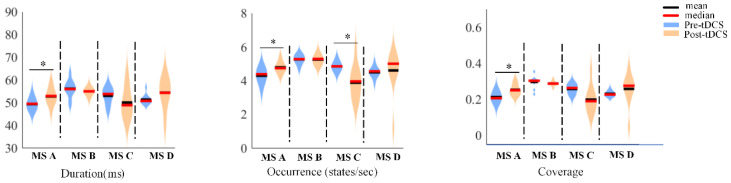
The differences in the three temporal parameters of the four microstates before tDCS and after tDCS for the experimental group. * denotes *p* < 0.05.

**Figure 6 bioengineering-10-00098-f006:**
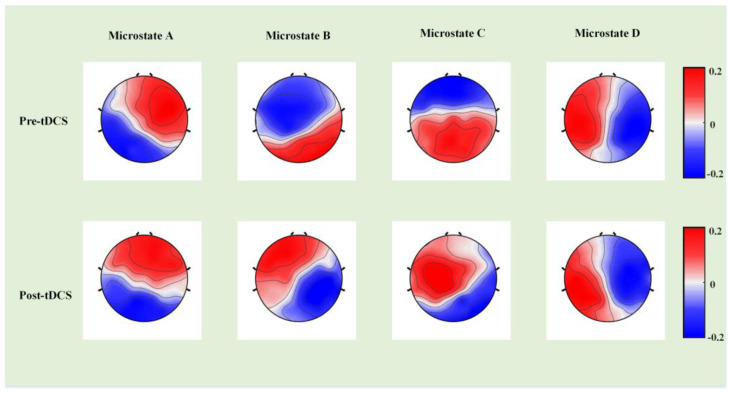
The template maps of the four microstate classes in the control group before and after tDCS.

**Figure 7 bioengineering-10-00098-f007:**
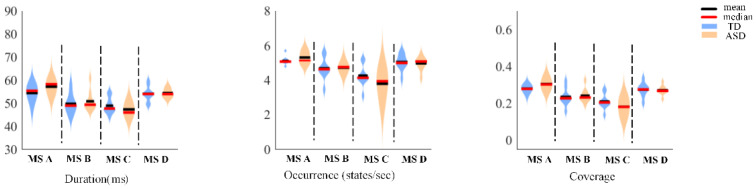
The differences in the three temporal parameters of the four microstates before tDCS and after tDCS for the control group.

**Table 1 bioengineering-10-00098-t001:** The scores of the ABC scale in the experimental group before and after tDCS.

ABC Scale	Pre-tDCS (Mean ± SD)	Post-tDCS (Mean ± SD)	*p*-Value	*z*-Value
S R B L S	13.31 ± 6.76 16.13 ± 4.52 14.45 ± 3.08 13.75 ± 5.50 11.88 ± 4.01	11.94 ± 6.38 11.80 ± 4.89 11.91 ± 3.08 9.31 ± 4.72 11.13 ± 3.76	0.12 0.002 * 0.113 0.019 * 0.589	1.555 3.090 1.585 2.346 0.540

(* *p* < 0.05; rest are not significant) Note: S: sensory behavior; R: social relating; B: body and object use; L: language and communication skills; S: social and adaptive skills.

## Data Availability

The datasets generated and analyzed in this study are available from the corresponding author upon reasonable request.

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
