# Peer review of "Transcranial Direct Current Stimulation Modulates EEG Microstates in Low-Functioning Autism: A Pilot Study"

_bioengineering, 2023, doi:10.3390/bioengineering10010098_

Round 1

Reviewer 1 Report

The sample size is very low. 

In the present study, 26 autistic children were recruited and were further divided into two groups: a tDCS treatment (experimental) and a sham stimulation (control) group with 13 each. This small size of the sample can undermine the validity of a study.  I suggest extending the number of samples.

Reviewer 2 Report

The article “Transcranial Direct Current Stimulation Modulates EEG Microstates in Low-Functioning Autism: A Pilot Study” presents results of experiments using tDCS in children with ASD, showing changes in the brain activity before and after the procedure. Authors compared brain activities of ASD children with children with typical development, finding that they have differences in some aspects. They also compared cerebral activity using EEG Microstare before and after experiments using really tDCS for 20 min with sham experiments where tDCS current was zeroed after 30 s, finding that electrical stimulation changes the activity of ASD children showing promises results in the treatment of these children.

Article is well written and easy to read, presenting sections in a good way. In the following are presented some points to complete or correct in order to have a better version.

- At lines 50 and 51, references 8 and 9 are cited as examples where Skinner’s model of learning was used, but it does not seem correct.

“Although the literature indicates that treatment based on Skinner’s model of learning can improve ASD, no fully effective treatments have been developed [8,9].”

8. Maras, A.; Schroder, C.M.; Malow, B.A.; Findling, R.L.; Breddy, J.; Nir, T.; Shahmoon, S.; Zisapel, N.; Gringras, P. Long-Term Efficacy and Safety of Pediatric Prolonged-Release Melatonin for Insomnia in Children with Autism Spectrum Disorder. J Child Adolesc Psychopharmacol 2018, 28, 699-710, doi:10.1089/cap.2018.0020.

9. Nitsche, M.A.; Liebetanz, D.; Tergau, F.; Paulus, W. [Modulation of cortical excitability by transcranial direct current stimulation]. Nervenarzt 2002, 73, 332-335, doi:10.1007/s00115-002-1272-9.

- Insert a reference to Autism Treatment Evaluation Checklist (ATEC) at line 74.

- At line 77 the year (2020) was left as part of other format of reference citation using (Author, year) or Author (year).

- At lines 122 and 123, when explaining the study design, the last phase was described as:

“(3) post-tDCS 122 treatment evaluation, consisting of EEG analysis for all three groups and ABC scale eval-123 uation for the experimental and control groups.”

However, at lines 156 and 157, when talking about EEG acquisition, a contradiction is found:

“In the TD group, EEG was conducted only once to compare EEG microstates between children with ASD and those with TD.”

According to the results presented, EEG is acquired in the TD children only in the beginning of experiments.

- At lines 155 and 156, and lines 157 and 158, it is not clear when EEG acquisition takes place, if it is before any section and after all sections of tDCS, or before and after a tDCS section at the same day.

“In the experimental group and the control group, EEG was conducted two times, once before tDCS and once after tDCS.”

“EEG was conducted 20 min before tDCS and 20 min after tDCS in a sound- and electrically shielded room by trained staff, and the participants were awake and in a resting state with their eyes open.”

- At line 166, explain better how artifacts were removed. Did the same person remove the artifact in all signals? Can this remoting compromise results? Was he(she) following a specific procedure?

“Artifacts, including electromyogram (EMG), eye blink, and muscular artifacts, were manually removed.”

- At line 183, it is missing a reference citation for global field power (GFP).

- At line 189, it is missing a reference for atomize and agglomerate hierarchical clustering (AAHC) algorithm.

- Sentence below is missing references for cited studies. It is supposed for the authors point out which studies they are talking about.

More information regarding AAHC can be found in other studies.”

- It is necessary to change figure number. Figure 5 is cited first of all figures (at page 4) and positioned at page 5. Figure 1 and Figure 2 are cited at page 6. Figure 3 and Figure 4 are cited at page 7.

- In Figure 5, Figure 1 and Figure 3, it is necessary to add a legend of color since results are presented and reader cannot understand what values are represented by red, blue and white colors.

- In Figure 2 and Figure 4, the symbol * is not presented. By text, it may represent significant difference, but the information should be in the graph or in the figures subtitle.

- At line 212, in the section Results, what does CARTOOL mean? Define the acronym and insert a reference for.

- There is a repetition at line 214 “the the”.

- Caption of Table 1 is in a different page than table itself.

At line 284, the sentence is missing a reference citation.

“A previous study suggested that atypical perceptual processing ability was associated with the autistic phenotype.”

At lines 332 and 333, there is a repetition for “in the experimental group”.

“In the experimental group, tDCS displayed significant differences in EEG microstate and ABC scale between pre- and post-tDCS in the experimental group.”

 At the discussion, it is necessary to explain better the consequences in EEG signals with eyes-open, considering electrodes position and frequencies analyzed and comparing with EEG signals acquired with eyes-closed. Why can eyes-open EEG acquisition ruin the results?

Finally, it is necessary to insert a space before each reference citation, separating square brackets from the last word before.

Reviewer 3 Report

The present study investigated the potential effects of tDCS on ASD. we analyzed EEG microstates and the Autism Behavior Checklist (ABC) scores to examine the potential therapeutic effects of tDCS. Microstates A, B, and D differed significantly between children with TD and those with ASD. In the experimental group, tDCS displayed significant differences in EEG microstate and ABC scale between pre- and post-tDCS in the experimental group. Conversely, in the control group, neither the EEG microstates nor the ABC scores before the treatment period (sham stimulation) differed from those after the treatment period. The present study indicated that tDCS may become a viable treatment for ASD.

I think the manuscript includes new and intriguing findings, however the authors should revie it according to the following concerns;

1.The authors should discuss on the ways to examine effects of tDCS on ASD other than EEG microstates and the superiority of EEE microstates over them.

2.The authors should discuss on the mechanism how tDCS causes beneficial effects of psychiatric symptoms of ASD and the reason why the beneficial effects of tDCS were correlated with the changes of EEG microstates in the present study.

Reviewer 4 Report

The number of figures is a bit messy. The authors should number figures according to the order in which they appear in the manuscript. For example, Figure 5 appears firstly, it should be compiled as Figure 1.

The authors should list the general clinical characteristics of the subjects in the table, such as age, sex, course of disease, presence or absence of complications and medication which may affect the outcome..

The authors should display the results compared with the sham group.

Reviewer 5 Report

First, I would like to thank you for the opportunity to review this study, I found it interesting to read about it and review the literature.

In this article the impact of active vs sham Transcranial direct current stimulation (tDCS) on electroencephalogram (EEG) microstates and Autism Behavior Checklist (ABC) scores were compared between children with typical development and those with ASD was evaluated.  Microstates A, B, and D differed significantly between children with TD and those with ASD. In the experimental group, microstates A and C and ABC scores before tDCS differed from those after tDCS. Conversely, in the control group, neither the EEG microstates nor the ABC scores before the treatment period (sham stimulation) differed from those after the treatment period.

I would like to congratulate the team that worked on the scientific article, and at the same time I would like to expose some situations that I have considered cannot be skipped.

Minor issues:

·      Please add the mean age of the 3 experimental groups.

·      Were there gender differences by groups?

·      Justify why you have chosen the intensity of 1 mA and not 2 mA. There is a lot of literature that indicates how the intensity is dependent on the effects that are generated. For example, you can check this:

Ammann, C., Lindquist, M. A., & Celnik, P. A. (2017). Response variability of different anodal transcranial direct current stimulation intensities across multiple sessions. Brain stimulation10(4), 757-763.

·      What solution can you think of for a future investigation to collect resting state data with eyes closed with children?

·      I recommend adding the following bibliographical references, since they could improve the article and help you justify the use of tDCS with people with autism;

Padrón, I., García-Marco, E., Moreno, I., Birba, A., Silvestri, V., León, I., Álvarez, C., López, J., & de Vega, M. (2022). Multisession Anodal tDCS on the Right Temporo-Parietal Junction Improves Mentalizing Processes in Adults with Autistic Traits. Brain Sci., 12 (1), 30. https://doi.org/10.3390/brainsci12010030 

De Vega, M., Padrón, I., Moreno, I., García-Marco, E., Domínguez, A., Marrero, H. & Hernández, S. (2019). Both the mirror and the affordance systems might be impaired in adults with high autistic traits. Evidence from EEG mu and beta rhythms. Autism Research, 12 (7), 1032-1042.  Doi: 10.1002/aur.2121.

For these reasons I consider its publication to Accept after minor revision.

Kind regards,

Round 2

Reviewer 1 Report

Accepted in present form 

Reviewer 4 Report

The author replied to all the questions.